# My Amyotrophic Lateral Sclerosis (ALS) Journey from Weakness to Diagnosis: A Journey of Hope

**DOI:** 10.3390/healthcare13212754

**Published:** 2025-10-30

**Authors:** Sherry Wityshyn, Nitesh Sanghai, Geoffrey K. Tranmer

**Affiliations:** 1University of Winnipeg, 515 Portage Ave., Winnipeg, MB R3B 2E9, Canada; 2College of Pharmacy, Rady Faculty of Health Sciences, University of Manitoba, Winnipeg, MB R3E 0T5, Canada; geoffrey.tranmer@umanitoba.ca

**Keywords:** Amyotrophic lateral sclerosis (ALS), genetic testing, community engagement, patient experience, edaravone, riluzole, policy makers, Canadian health, drug discovery and development

## Abstract

**Highlights:**

**Genetic analysis is still a challenge in ALS. It should be recommended during diagnosis, leading to early inclusion of patients in the clinical trials.**

**Eligibility criteria in ALS clinical trials are strict. Refining inclusion criteria, including patients from all clinical stages of progression, is needed.**

**Canadian health care providers provide meaningful support to ALS patients.**

**The drug discovery and development pipeline for creating effective ALS therapies should involve a wider collaborative community, including patients, caregivers, basic scientists, clinicians, and medicinal chemists.**

**Funding research is just as crucial as funding healthcare by the government. Innovations from laboratories turn into real health solutions. Therefore, supporting research in ALS is essential.**

**Abstract:**

Amyotrophic lateral sclerosis (ALS) or Lou Gehrig’s disease is a progressive neurodegenerative disease that attacks and kills motor neurons in the brain and spinal cord, leading to muscle weakness and atrophy, eventually causing respiratory failure and death within 2–5 years after diagnosis. By 2040, the global population of individuals living with ALS is projected to approach 400,000. Since ALS was discovered by Charcot 150 years ago, only two drugs (Edaravone and Riluzole) have been available, offering modest clinical benefits in slowing disease progression. The increasing number of cases, along with the high costs of treatment and care, creates a growing burden on communities and the healthcare system. However, despite this rising burden and the failure of most clinical trials, the ALS community remains hopeful because of the patients themselves. ALS patients are the beating heart of the ALS community. They engage in efforts to improve lives for others, raising awareness through their real-life experiences, participating in research activities, fundraising, providing samples for research, and advocating strongly in front of communities and governments to raise funds. Their engagement is highly valuable, and collaboration with the research community is essential to understanding the disease process and developing effective disease-modifying therapies. Here, we share the story of Mrs. Sherry Wityshyn, an ALS patient and a true ALS warrior from Winnipeg, Manitoba, Canada. We believe her story will inspire and motivate the entire community to learn more about ALS. Furthermore, her story gives hope to everyone impacted. In this manuscript, we also emphasize the different stages of Sherry’s journey from weakness to diagnosis and our efforts to share her enduring words with policymakers in the government.

## 1. Introduction

Amyotrophic lateral sclerosis (ALS) is a fatal motor neurodegenerative disease [1]. The worldwide number of ALS cases is expected to grow from 222,801 in 2015 to 376,674 in 2040, a 69% increase [2]. ALS was first identified 150 years ago by French neurologist Jean-Martin Charcot [3]. Since then, the scientific community has not determined its cause, and no cure is available. Today, the disease is a major burden, with a significant need for new drug development. Patients with ALS are central to ALS research; their experiences, participation in clinical trials, and data sharing are vital for understanding and treating the disease. Research relies heavily on patient involvement, from detailed medical histories to participation in clinical trials and therapeutic testing. Patients also play a key role in fundraising efforts, such as the 2014 viral Ice Bucket Challenge, which significantly increased ALS research funding and contributed to advances in understanding and treatment. Additionally, patient advocates have played a crucial role in the recent approval of Tofersen, an antisense oligonucleotide (ASO), for the treatment of ALS associated with a mutation in the superoxide dismutase 1 (*SOD1*) gene. Consequently, patients play a vital role in understanding the disease’s pathology. Here, we share a story written by Sherry Wityshyn, an ALS patient and a true ALS warrior from Winnipeg, Manitoba, Canada. We believe her story will inspire and motivate the entire community to learn more about ALS. Moreover, her story will bring hope to the community [4].

The story below was written by Sherry Wityshyn with the help of Eyegaze (Tobii Dynavox Eyegaze computer).

## 2. Aim and Purpose of This Story

ALS is a highly complex, multifactorial, idiopathic, heterogeneous, and incurable rare motor neurodegenerative disease. People worldwide are conducting research to slow the progression and improve the quality of life for ALS patients. For years, researchers have been making every effort to push the drug discovery engine to find a silver bullet molecule that could potentially slow disease progression. However, even 150 years after Charcot’s discovery of ALS and 30 years after the identification of SOD1 mutations, progress in discovering and developing new treatments remains limited. Nevertheless, ALS patients and caregivers continuously inspire and motivate both the ALS and scientific communities. Despite suffering from ALS, patients continue to fight for the community and to raise funds for research worldwide. Additionally, ALS is a heterogeneous disease, with each patient experiencing different rates of progression, which makes understanding it highly complex [5]. Many ALS patients may lose hope due to the disease’s progressive and debilitating nature, which is caused by several factors. First, diagnostic delays often mean patients receive a diagnosis too late, sometimes because the disease progresses rapidly [6]. Overlapping symptoms with other neurological disorders can further delay diagnosis. Second, resource scarcity poses a challenge, including financial burdens from expensive treatments, limited access to assistive devices like accessible vans, inadequate in-home care, and gaps in caregiver support [7]. Third, navigating clinical trials is difficult for ALS patients due to logistical barriers, strict eligibility criteria, and informational challenges [8]. Fourth, barriers to genetic testing include inconsistent access, high costs, limited genetic counseling, and provider awareness issues [9]. Lastly, many patients lose hope in the drug discovery process itself, as most clinical trials have failed to demonstrate efficacy, potentially due to the disease’s complexity and heterogeneous nature, as well as ill-defined clinical endpoints and poorly designed clinical trials [10]. However, we firmly believe that the drug discovery pipeline for ALS should involve a broader scientific community, including patients, basic scientists, clinicians, and medicinal chemists. Currently, medicinal chemists are underrepresented in the discovery and development of ALS drugs. They develop drugs by designing, synthesizing, and optimizing chemical compounds to create new medicines [11].

We strongly believe that sharing patient stories can raise awareness about the disease and foster strong advocacy within the ALS community for specialized, holistic, multidisciplinary care [12]. Sherry’s story is unique. Sherry’s journey from weakness to diagnosis highlights the diagnostic challenges, advocates for genetic testing, and showcases her resilience, while also emphasizing the importance of flexible clinical trial inclusion criteria. Through her story, Sherry remains highly optimistic about improving the lives of ALS patients by raising awareness, encouraging resilience, perseverance, and fostering hope. Her story aims to influence the perspectives of ALS patients and inspire positive change. Additionally, her story aims to motivate, inspire, and encourage ALS patients worldwide to develop a positive will to live despite the disease.

My name is Sherry. I was diagnosed with amyotrophic lateral sclerosis (ALS) on 18 November 2015, at the age of 47 (Figure 1). I grew up in Winnipeg before moving to Île-des-Chênes for my teenage years. Summers were spent at our family cabin in Albert Beach. I have three siblings: an older brother and two younger sisters. My diagnosis was a shock to them because we’re close, and it led us to wonder if it might be genetic or related to environmental factors like pesticides or chemicals. I was a competitive swimmer and active through much of my life. I have four children; they were young when I was diagnosed, with the youngest being 11. I held a desk job and earned Certified Employee Benefit Specialist (CEBS) and Certified Human Resources Professional (CHRP) designations, which made my busy, joyful life possible. I am a lifelong learner, passionate about reading and research. Our family is familiar with rare diseases. My first grandchild was diagnosed with an atypical Teratoid/Rhabdoid Tumor (ATRT) brain tumor at the age of one and passed away a year later. During this time, my husband was diagnosed with congestive heart failure at the age of 44.

I was very in tune with my body, so I noticed symptoms of weakness early on. In March 2015, my right-hand pointer and index fingers felt numb and tingly. My chiropractor suggested it might be due to a fall at the end of January 2015 when I slipped on ice, falling on my right hip and shoulder, and hitting my chin. I even loosened my front lower teeth. During February and March, I experienced constant shoulder and hip pain, making it hard to sleep on my right side. My hip pain worsened with movement, and my fingers grew weaker, prompting me to see my doctor. In April, I was advised to try anti-inflammatory medication, and I kept seeing my chiropractor, who urged me to undergo more laboratory tests. By late June, I struggled with cursive writing and kept dropping my pen, and I also had trouble clicking my mouse. My doctor then referred me to a sports medicine specialist for my hip and an electromyography (EMG) neurologist for possible carpal tunnel syndrome. I was seen quickly by both specialists. The sports doctor considered a cortisone shot, but I hesitated due to my daughter’s upcoming wedding. The nerve conduction study was normal; however, the EMG test showed slight abnormalities. A computed tomography (CT) scan of my cervical spine was normal, and I was referred to physiotherapy. Despite intensive physiotherapy starting in August, my condition worsened; my right hand weakened further. By mid-August, fasciculations appeared in my right hand and arm. On the 19th, I saw the EMG doctor again, who ordered a Magnetic Resonance Imaging (MRI) of my shoulder’s brachial plexus and cervical spine, but there was nothing conclusive. I began researching online, and the findings worried me. I continued physiotherapy and chiropractic care, but no improvement occurred. Further, I noticed muscle atrophy between my thumb and index finger, and could no longer sign my full name or use hair clips or spray bottles. By September, fasciculations appeared in my left arm too, though I felt no weakness. I saw the EMG doctor again at the end of September; the MRIs showed no issues, but the presence of fasciculations in both arms increased concern. The EMG doctor explained various possibilities, including immune problems, but worst-case, ALS. That was a challenging diagnosis to accept. By early October, my weakness had increased, with more widespread fasciculations. I was referred to a neuro-musculoskeletal specialist at the Health Science Centre (HSC) and saw her on 29 October. After a thorough neurological examination, the doctor expressed her probable suspicion of ALS. However, further tests such as blood work for immune issues, acquired immunodeficiency syndrome (AIDS), syphilis, heavy metals, and others were necessary. Finally, on November 18th, I received the diagnosis I feared most: ALS. I later sought a second opinion from doctors, but they also confirmed it was ALS. There never appeared to be any other diagnosis they considered. It was a straightforward diagnosis once they had ruled out ALS mimics and other overlapping neurodegenerative diseases (NDDs) (Figure 2).

It is worth mentioning here that during my health examination, all the doctors showed great interest in my medical condition and were extremely compassionate and supportive. I was diagnosed quite quickly, I think within a year, but certainly after several months. After my definitive ALS diagnosis, I kept working for around four months, but it eventually became too hard as both my hands and speech were affected. I’m part of the Canadian Longitudinal Study on Aging, which did my whole genome sequencing but didn’t share the results with me. So, I also did tests with Ancestry and 23andMe. I wanted to do this to help my siblings and children, in case questions about genes arise after I’m gone, and to get the raw data. I discovered this by accident through 23andMe before Canadian regulations limited what they share. I made an interesting discovery: I am homozygous (+/+) for the *Delta 32* mutation, a rare variant found in less than 1% of people. Because of this, my chances of contracting human immunodeficiency virus (HIV) are very low or nonexistent. I also asked my siblings to test their genetics, and each of them has one copy. Although this doesn’t appear to be connected to ALS, I’m uncertain. Additionally, I used Genetic Genie to assess methylation. Through this DNA analysis, I learned that I have mutations in folate and vitamin D genes: “VDR BsmI” and serine hydroxymethyltransferase (*SHMT1)* C1420T (+/+), the genotype related to vitamin D receptor function and folate metabolism. In addition, I also tested for the common ALS genes *SOD1* and *C9orf72*, but not TAR DNA-binding protein 43 (*TARDBP)* and fused in Sarcoma (*FUS*), since they weren’t being tested in Canada at the time. My father’s Frontotemporal dementia (FTD) prompted the tests. Doctors told me I don’t have those specific mutations, but I would like a geneticist to review other known genes, as I’ve only researched this myself. My siblings also did 23andMe, and I told them where to look.

I have been taking Riluzole since my diagnosis and compounded Dextromethorphan/quinidine (Nuedexta) since 2018, when I developed Pseudobulbar Affect. I also take Amitriptyline and Nabilone for saliva management and sleep. Additionally, I have B12 shots twice a week following the results of a study, posted on social media. I also take several supplements, including all the B vitamins, C, and D, along with magnesium, melatonin, milk thistle, and probiotics. I used to take cannabis oil before Nabilone. Before ALS, I didn’t take any prescribed or recreational medications. The medications doctors try for saliva have major side effects on other physiological processes; I have tried them all and know firsthand that being too dry isn’t good either. So, I use a bite cloth in my mouth to absorb extra saliva, as shown in my picture above. It works well for me without causing dryness, which leads to constipation. However, my jaw strength is extraordinary, so strong that it has shifted my teeth all over the place. I never thought I’d miss my smile so much! Aside from that, I experience minor aches and pains from stretching my tendons and ligaments. Overall, I focus on comfort since I can’t move much, and I feel I have that covered.

I wish I could help researchers more. However, the inclusion criteria for participating in the clinical trials are challenging. The investigator usually limits trials to under two years since diagnosis and a specific “The Revised Amyotrophic Lateral Sclerosis Functional Rating Scale” (ALSFRS-R) score. I still move quite a bit, even if passively. Furthermore, I can easily tell if a drug is working. I want to help the scientific community with my case of confirmed ALS, but no clinical studies currently accept me. I’m willing to give blood and have already had my DNA sequenced; it’s all here waiting for research. I also hope doctors will investigate liver issues and nutrition. I don’t use formula in my feeding tube; instead, I have good, nutritious food in the right amounts, which supports my health. I couldn’t wait for the feeding tube to eat healthier, especially when taste isn’t an issue. In addition, I maintain a steady weight and have never lost weight, which is essential. I’ve been living with this for nearly 10 years and fast for 12 h each day. In the past, I ate healthily too. When I was diagnosed, I visited the Motor Neuron clinic every three months, but now I go annually since I haven’t significantly progressed in over three years. I also see my family doctor once a year. I receive maximum hours of home care from the government, and I pay for some services. I have caregivers, either family or staff, around the clock. I can’t be alone ever, which is challenging for an introvert. I sleep through the night and am flipped once. Placing an ALS patient in a nursing home isn’t ideal because they aren’t skilled in this type of care. With this disease and the associated symptoms, when given 3–5 years to live, every extra day feels like a gift! I want to see my family, kids, and grandkids; they are what truly keep me going! I‘m okay with changes; they challenge me to always look ahead. I believe I’ve experienced tremendous personal growth, and I now have more time for it, which I didn’t have before.

Any advice is beneficial to ALS patients, but each person progresses at their own specific pace. Each individual experiences different symptoms depending on where the disease manifests in a muscle. There are common issues that affect us all, but you won’t hear about them from a doctor. There are often simple solutions. For example, complaints of acid eyes, burning, and stinging are very common and painful, but a simple fix is washing your eyelashes when you wake up until all the accumulated tear salt is gone. My eye doctor told me it’s dry eye, which seems to be common among all patients. I was advised to use drops daily, which I do, but he never mentioned washing lashes. It’s such a simple solution that many doctors might not know about. I also worry about choking all the time, even when food is blended or you are tube-fed. How can I choke on nothing? These episodes may seem like choking, but actually, it’s your body’s defense mechanism against choking. The vocal cords slam shut to prevent anything from entering the windpipe. This is called a laryngeal spasm. When I understood this, it changed my perspective; instead of feeling like I’m constantly choking, I realized my body is just doing its job—protecting me. Laryngeal spasms can be frightening, but you will hear a sound like a seal when you breathe in. They usually pass in a few minutes. These spasms can be triggered by postnasal drip or a tiny bit of food, like a crumb. Sometimes they happen at night, waking you up because you can’t breathe.

The last thing I want to mention is that good nutrition and staying hydrated are important! My doctor told me to eat whatever I want while I can, and I initially ate a lot of Häagen-Dazs ice cream. Now I know better. When you have enough fluids, your mucus stays thin, and eating nutritious foods that provide maximum fuel for your cells will help reduce tiredness. I eliminated all unnatural sugars, processed foods, and all dairy unless it was fermented, like yogurt. It was easy because of my feeding tube! Don‘t be afraid of a feeding tube; it’s the easiest and safest way to prevent choking. Being a Canadian with this disease is very different from other countries. We receive benefits that others do not; many are full-time caregivers for loved ones with ALS, balancing the costs of needed items with living expenses. It’s often a constant struggle. Here in Canada, most jobs offer benefits like drug coverage and long-term disability. We also have government support through CPP disability, and we can access sick leave from the government. Family members can also take caregiver leave. Most major cities have ALS/MND clinics with all the specialists you need on this journey. In Manitoba, they are very well-organized, from home care to supplies. Most things in Manitoba are easy to access with short wait times. The Motor Neuron Disease (MND) Clinic and the ALS Society of Manitoba are excellent at supporting us, always knowing about resources we need before we realize it. Canada’s disability laws make things easier for people with this disease. There are many options and accommodations for people with disabilities. All I’m saying is we were dealt a tough hand, but where we live makes a big difference. We have freedoms that others don’t and social services that help in many ways. I am a proud Canadian, and I couldn’t be more thankful. Grateful for my life and the worries I leave in the hands of our creator, God!

## 3. Conclusions

Sherry’s conclusion and wish, conveyed through her story of resilience, is to establish an ALS research hub dedicated to improving the early diagnosis of the disease. This hub should include genetic counseling and testing for all patients, regardless of their diagnosis, and encompass their family members as well. Additionally, she advocates for a centralized, user-friendly knowledge portal for ALS patients and caregivers, providing current information on ongoing clinical trials, with clear navigation and criteria. She also urges broader inclusion criteria for clinical trial participation, emphasizing that ALS, being a rare and incurable disease, warrants that all patients should have the right to participate, regardless of disease stage, genetic makeup, or heterogeneity. She believes every patient’s contribution can advance drug discovery and development. Furthermore, Sherry emphasizes the importance of diverse research teams, including clinicians, scientists, medicinal chemists, patients, and caregivers, to incorporate multiple perspectives in the drug discovery process. Through her story, she aims to advocate for increased government funding for ALS research.

## 4. Final Words for the Policy Makers in the Government

Rare diseases, such as ALS, can often become neglected diseases, as the numbers of patients seem small, but the need remains high. Policy makers in the government need to focus on the high need of ALS patients for inclusion in the latest research. Serving the high needs of one ALS patient serves the needs of many, including caregivers, the community, and practitioners.

## Figures and Tables

**Figure 1 healthcare-13-02754-f001:**
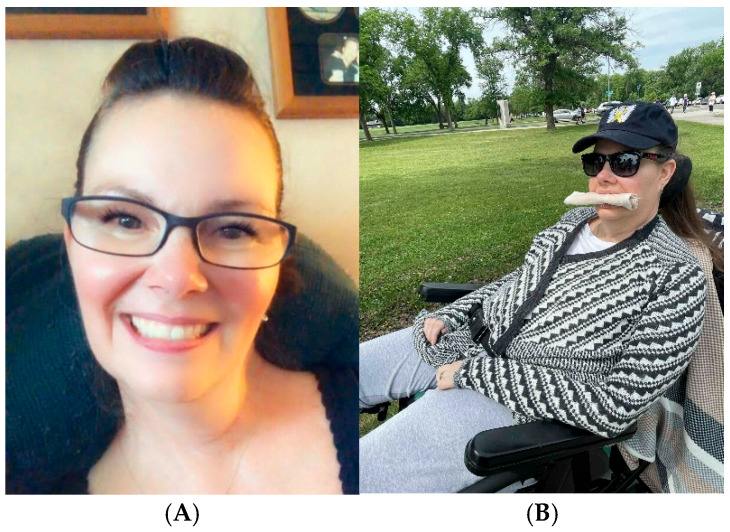
(**A**) Picture on the left (Taken after diagnosis in 2022, almost 8 years after my diagnosis). The photo was taken in 2022 during Mother’s Day; all my family were celebrating with me that day. All four of our kids and grandkids were over. I was using my AAC Tobii Dynavox Eye Gaze computer to chat with all of them. I don’t know what would happen if I didn’t have this communication device. It keeps me connected with everyone, by text, email, social media, and in-person communication. I am truly grateful they have this type of communication device; (**B**) Picture on the right (7 June 2025, almost 10 years after my diagnosis). I participated in the annual ALS walk 2025 (walk to end ALS), a fundraising event organized by the ALS Society of Manitoba (ALS MB), with the mission of creating “Hope in the ALS community”. This family-friendly event promotes unity among Canadians, all sharing a common goal: to eradicate ALS. It was a beautiful day, surrounded by friends and family. I’m so happy that I’m still able to get out and about. MB Possible Supply wheelchairs (funded by the government of Manitoba) for all who need them in Manitoba. The ALS MB supplies seat cushions and backrests. I’m so Grateful to have these organizations to help with expensive and cost-prohibitive items. And to have people to donate to these wonderful organizations.

**Figure 2 healthcare-13-02754-f002:**
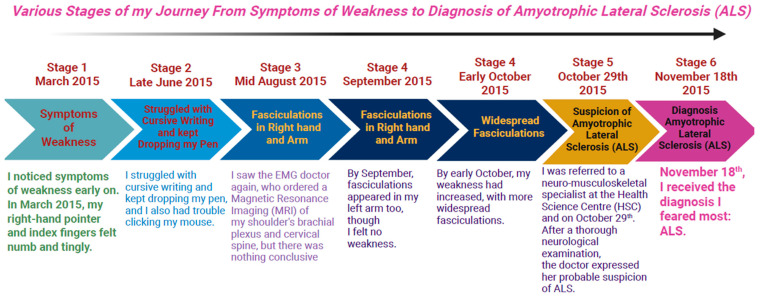
Various Stages of my Journey from Symptoms of Weakness to Diagnosis of Amyotrophic Lateral Sclerosis (ALS).

## Data Availability

No new data were created or analyzed in this study.

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
