# Peer review of "My Amyotrophic Lateral Sclerosis (ALS) Journey from Weakness to Diagnosis: A Journey of Hope"

_healthcare, 2025, doi:10.3390/healthcare13212754_

Round 1

Reviewer 1 Report

Comments and Suggestions for Authors

This Perspective represents a personal reflection from an ALS patient, developed in collaboration with two researchers. The manuscript does not pose a central research question; rather, its primary aim is to draw attention to how individuals living with ALS can actively engage in and contribute to the advancement of ALS research. In this regard, the topic is both original and relevant to the field, as it bridges the gap between patients seeking involvement in research and scientists striving to develop more effective treatments and cures. There are relatively few publications offering perspectives from patients themselves, and thus this work provides valuable insights into fostering collaboration among all stakeholders in the ALS community—including patients, clinicians, researchers, caregivers, and policymakers—when compared with existing literature. Given the absence of a specific research question, definitive conclusions cannot be drawn. The references cited are appropriate. Overall, I believe this manuscript merits open access publication in Healthcare, ensuring that it is freely available to the public. I have no major criticisms, only the following three minor suggestions. First, in line 107, “VDR Bsm” should be corrected to “VDR BsmI.” Second, gene symbols should be italicized to preserve scientific accuracy. Finally, the figure should include an appropriate title and descriptive legend.

Author Response

Reviewer 1

Does the introduction provide sufficient background and include all relevant references?

Yes

Is the research design appropriate?

Yes

Are all figures and tables clear and well-presented?

Yes

Comments and Suggestions for Authors

This Perspective represents a personal reflection from an ALS patient, developed in collaboration with two researchers. The manuscript does not pose a central research question; rather, its primary aim is to draw attention to how individuals living with ALS can actively engage in and contribute to the advancement of ALS research. In this regard, the topic is both original and relevant to the field, as it bridges the gap between patients seeking involvement in research and scientists striving to develop more effective treatments and cures. There are relatively few publications offering perspectives from patients themselves, and thus this work provides valuable insights into fostering collaboration among all stakeholders in the ALS community—including patients, clinicians, researchers, caregivers, and policymakers—when compared with existing literature. Given the absence of a specific research question, definitive conclusions cannot be drawn. The references cited are appropriate. Overall, I believe this manuscript merits open access publication in Healthcare, ensuring that it is freely available to the public. I have no major criticisms, only the following three minor suggestions. First, in line 107, “VDR Bsm” should be corrected to “VDR BsmI.” Second, gene symbols should be italicized to preserve scientific accuracy. Finally, the figure should include an appropriate title and descriptive legend.

NS-GKT Comments: Dear Reviewer 1, thank you for reviewing the manuscript. I appreciate the time and effort you took to critique it thoughtfully. Additionally, I want to thank you for your kind words about the manuscript. This is the first time our group under Dr. Tranmer has decided to publish the patient’s perspective. In addition to our research, our group is actively advocating for and engaging with ALS patients in the community. Our primary intention with this patient’s story is to create “HOPE” in the lives of those suffering from ALS. Furthermore, we aim to raise awareness and strongly advocate for ALS, a highly complex and incurable motor neurodegenerative disease.

We have now addressed the comments in the track change in the main manuscript. We have changed “VDR Bsm to “VDR BsmI.” And italicized all the gene symbols throughout the manuscript. Further, we have given a caption for Figure 1. Additionally, we have introduced a new Figure 2, which illustrates various Stages of my Journey from Symptoms of Weakness to Diagnosis of Amyotrophic Lateral Sclerosis (ALS). This figure is unique, and we tried to simplify the journey of Mrs. Sherry through this figure, mentioning the journey in different stages. Please see Figure 2, which we have included in the manuscript.

Reviewer 2 Report

Comments and Suggestions for Authors

Opening context (“ALS is a fatal motor neurodegenerative disease … we share a story…”)

  • The framing is fine, but major assertions (e.g., Ice Bucket Challenge funding impact; advocacy role in tofersen approval) should be paired with citations or softened to “widely credited with…”. Right now, only four references appear and none cover those claims. Add 2–4 targeted citations or qualify the wording.
  • Keep “inspire and motivate” but avoid promising outcomes (“will bring hope to the whole community”), which reads absolute.
  • There’s no formal abstract. Even for a Perspective, include a brief structured or unstructured abstract (3–5 sentences) summarizing the narrative arc and the 2–3 take-home points (genetics access, trial eligibility, care coordination).

Patient narrative (onset → diagnostic work-up)

  • “pointer and index fingers” → redundant; use “index finger.”
  • Consider a pared-down timeline table (month/year | symptom/test | result/impact). It will help reviewers follow the sequence (fall → EMG → MRI → referral → diagnosis).
  • Reported tests are “normal” or “slightly abnormal.” If available, add the key positives/negatives (e.g., UMN/LMN signs on exam, EMG distribution) to help readers understand why ALS became most likely after mimics were excluded. If details aren’t available, a brief note about limited access to reports keeps transparency.
  • Avoid declaring it was a “straightforward diagnosis” after ruling out mimics; for many patients it’s nuanced. Consider “the diagnosis became likely once ALS mimics were excluded.”

Genetics section (consumer genomics, CCR5-Δ32, VDR/SHMT1, SOD1/C9orf72)

  • Specify CCR5-Δ32 rather than “Delta 32 mutation,” and clarify that homozygosity is typically discussed in the context of HIV susceptibility—not known ALS causality. Emphasize that direct-to-consumer findings (VDR Bsm, SHMT1 C1420T) have uncertain clinical significance for ALS and shouldn’t be over-interpreted. Suggest adding a line recommending genetic counseling and, if feasible, clinical-grade ALS panels (including TARDBP/FUS which you note weren’t available at the time).
  • Add at least one citation on (a) the prevalence and typical implications of CCR5-Δ32 and (b) the current role/limits of genetics in ALS care. If opting not to add, explicitly frame as personal exploration to avoid implying causality.
  • Mentioning the Canadian Longitudinal Study on Aging (WGS not returned) is valuable. Briefly add consent/data-return context (e.g., “research-only sequencing with no individual return of results”), or remove the policy critique tone unless you can reference program policy.

Treatment & symptom management (riluzole, Nuedexta, amitriptyline, nabilone, B12, supplements)

  • Use generic names first (e.g., dextromethorphan/quinidine for Nuedexta). “Compounded Nuedexta” may need a clarifying phrase about access/regulatory status; otherwise readers could misconstrue.
  • The B12 twice weekly “as part of a research study” should either include a brief descriptor (study type/ID/aims) or be pared down; currently it reads like an efficacy suggestion without context. Same for supplements—frame as personal choices; avoid implying disease-modifying effects. Add a one-sentence safety note (interactions/side-effects) to model responsible patient-authored guidance.
  • Pseudobulbar affect & sialorrhea
    Great to include lived tips (bite cloth). Consider adding 1–2 clinical pointers (“discuss with your care team …”) and avoid prescriptive language.

Clinical trials & eligibility

  • The scale is ALSFRS-R, not “ALSFR-R/ALSFR-R.” Please correct throughout. This is a high-salience error for reviewers.
  • The critique of strict eligibility is reasonable, but it needs either citations (platform trials, expanded access, adaptive designs) or reframing as a personal wish. A suggested constructive line: “Broader, phase-appropriate inclusion or pragmatic trials could enable participation across stages while still preserving safety and interpretability.”
  • “I can easily tell if a drug is working” invites reviewer pushback (placebo, regression to mean). Soften: “I’m highly attuned to changes and would value contributing patient-reported outcomes to trials.”

Practical tips (eyes/lashes, laryngospasm)

  • These are useful. To avoid over-generalization: add “This reflects my experience; please consult clinicians for diagnosis/treatment.”
  • “Laryngeal spasm” → “laryngospasm” is the more common clinical term; add a brief, neutral description. If you keep it, a single citation would go a long way; if not adding refs, explicitly flag as experience-based.

Nutrition/hydration & fasting

  • The nutrition section reads prescriptive (“Don’t be afraid of a feeding tube”; “eliminated all unnatural sugars”). Reframe as “what worked for me,” and encourage dietitian involvement. Consider trimming brand callouts (e.g., “Häagen-Dazs”) to avoid perceived product placement.
  • Statements implying fatigue reduction or mucus changes from diet should be hedged unless you will cite. Otherwise: “I felt less tired when…” keeps it appropriately subjective.

Health-system reflections (Canada/Manitoba)

  • Excellent contextualization, but claims about “most jobs” and “most cities” should be narrowed (“In my experience in Manitoba…”) or supported. If you keep the national view, add a qualifier noting variability by province/insurer.
  • A single sentence acknowledging that supports vary across regions and socioeconomic contexts will pre-empt reviewer critiques about representativeness.

References

  • Only four citations for wide-ranging claims is light, especially for the early context (Ice Bucket Challenge impact; advocacy and regulatory timeline; genetics at diagnosis; clinical-trial eligibility patterns; laryngospasm in ALS). Either (a) add 4–8 focused references, or (b) explicitly mark those statements as personal observations.
  • Ensure uniform journal style (punctuation, author initials, capitalization, DOIs). Verify “ALSFRS-R” appears correctly wherever the scale is mentioned (also in any reference titles if relevant).

Formatting & style housekeeping

  • Standardize dashes (en-dashes for number ranges), units, and capitalization (e.g., “Magnetic Resonance Imaging (MRI)” then “MRI” thereafter).
  • Spell out acronyms on first use (ALSFRS-R; AIDS already OK; EMG/MRI OK).
  • Tighten long sentences and remove colloquialisms that might read as prescriptive medical advice; keep the strong patient voice while adding light hedges (“in my case,” “I found”).

Author Response

Reviewer 2

Does the introduction provide sufficient background and include all relevant references?

Must be improved

Are all figures and tables clear and well-presented?

Must be improved

Comments and Suggestions for Authors

The framing is fine, but major assertions (e.g., Ice Bucket Challenge funding impact; advocacy role in tofersen approval) should be paired with citations or softened to “widely credited with…”. Right now, only four references appear and none cover those claims. Add 2–4 targeted citations or qualify the wording.

NS-GKT Comments: Dear Reviewer 2, thank you for reviewing the manuscript. I appreciate the time and effort you took to critique it thoughtfully. We have accepted your recommendations and suggestions, and therefore, we have now increased the citations from 4 to 12. These papers cited in the manuscript are in line with the aims and objectives of the manuscript.

Keep “inspire and motivate” but avoid promising outcomes (“will bring hope to the whole community”), which reads absolute.

NS-GKT Comments: Dear Reviewer 2, thank you for reviewing the manuscript. I appreciate the time and effort you took to critique it thoughtfully. We have now removed the word whole and included only hope to the community.

There’s no formal abstract. Even for a Perspective, include a brief structured or unstructured abstract (3–5 sentences) summarizing the narrative arc and the 2–3 take-home points (genetics access, trial eligibility, care coordination).

NS-GKT Comments: Dear Reviewer 2, thank you for reviewing the manuscript. I appreciate the time and effort you took to critique it thoughtfully. We have added two more key points in line with the patient's perspective.

Patient narrative (onset → diagnostic work-up)

NS-GKT Comments: Dear Reviewer 2, thank you for reviewing the manuscript. I appreciate the time and effort you took to critique it thoughtfully. We have accepted your recommendations and added Figure 2. Various Stages of my Journey from Symptoms of Weakness to Diagnosis of Amyotrophic Lateral Sclerosis (ALS).

“pointer and index fingers” → redundant; use “index finger.”

NS-GKT Comments: Dear Reviewer 2, thank you for reviewing the manuscript. I appreciate the time and effort you took to critique it thoughtfully. We have accepted your suggestions and removed the pointer.

Consider a pared-down timeline table (month/year | symptom/test | result/impact). It will help reviewers follow the sequence (fall → EMG → MRI → referral → diagnosis).

NS-GKT Comments: Dear Reviewer 2, thank you for reviewing the manuscript. I appreciate the time and effort you took to critique it thoughtfully. We believe that it is an excellent recommendation.

We have accepted your recommendations and added Figure 2. Various Stages of my Journey from Symptoms of Weakness to Diagnosis of Amyotrophic Lateral Sclerosis (ALS).

Reported tests are “normal” or “slightly abnormal.” If available, add the key positives/negatives (e.g., UMN/LMN signs on exam, EMG distribution) to help readers understand why ALS became most likely after mimics were excluded. If details aren’t available, a brief note about limited access to reports keeps transparency.

NS-GKT Comments: Dear Reviewer 2, thank you for reviewing the manuscript. I appreciate the time and effort you took to critique it thoughtfully.

However, this is beyond the scope of the manuscript. The patient is immobile and has both limb and bulbar onset. We cannot go into detail about all the clinical test results. We made every effort, taking several months to develop this story because of her progressive neurodegeneration. This is the best we could do.

Avoid declaring it was a “straightforward diagnosis” after ruling out mimics; for many patients it’s nuanced. Consider “the diagnosis became likely once ALS mimics were excluded.”

Genetics section (consumer genomics, CCR5-Δ32, VDR/SHMT1, SOD1/C9orf72)

NS-GKT Comments: Dear Reviewer 2, thank you for reviewing the manuscript. I appreciate the time and effort you took to critique it thoughtfully.

However, we have made every effort to bring the story in its original form. Because the patient herself wrote the story. Further, according to your suggestions, we have included Figure 2 for a better understanding of the diagnosis. Moreover, every ALS case is unique and diagnosed differently, depending on the availability of healthcare facilities.

Specify CCR5-Δ32 rather than “Delta 32 mutation,” and clarify that homozygosity is typically discussed in the context of HIV susceptibility—not known ALS causality. Emphasize that direct-to-consumer findings (VDR Bsm, SHMT1 C1420T) have uncertain clinical significance for ALS and shouldn’t be over-interpreted. Suggest adding a line recommending genetic counseling and, if feasible, clinical-grade ALS panels (including TARDBP/FUS which you note weren’t available at the time).

NS-GKT Comments: Dear Reviewer 2, thank you for reviewing the manuscript. I appreciate the time and effort you took to critique it thoughtfully.

However, this is beyond the scope of the manuscript. The patient is immobile and has both limb and bulbar onset. We cannot provide detailed information about all the clinical test results. We made every effort, taking several months to develop this story because of her progressive neurodegeneration. This is the best we could do.

Add at least one citation on (a) the prevalence and typical implications of CCR5-Δ32 and (b) the current role/limits of genetics in ALS care. If opting not to add, explicitly frame as personal exploration to avoid implying causality.

NS-GKT Comments: Dear Reviewer 2, thank you for reviewing the manuscript. I appreciate the time and effort you took to critique it thoughtfully.

However, we strongly believe that this is beyond the scope of the manuscript.

Mentioning the Canadian Longitudinal Study on Aging (WGS not returned) is valuable. Briefly add consent/data-return context (e.g., “research-only sequencing with no individual return of results”), or remove the policy critique tone unless you can reference program policy.

NS-GKT Comments: Dear Reviewer 2, thank you for reviewing the manuscript. I appreciate the time and effort you took to critique it thoughtfully.

However, we firmly believe that this is beyond the scope of the manuscript.

Treatment & symptom management (riluzole, Nuedexta, amitriptyline, nabilone, B12, supplements)

NS-GKT Comments: Dear Reviewer 2, thank you for reviewing the manuscript. I appreciate the time and effort you took to critique it thoughtfully.

This is what the patient has written about her treatment and symptom management.

Use generic names first (e.g., dextromethorphan/quinidine for Nuedexta). “Compounded Nuedexta” may need a clarifying phrase about access/regulatory status; otherwise readers could misconstrue.

NS-GKT Comments: Dear Reviewer 2, thank you for reviewing the manuscript. I appreciate the time and effort you took to critique it thoughtfully.

We have accepted your suggestions and made changes accordingly in the manuscript.

The B12 twice weekly “as part of a research study” should either include a brief descriptor (study type/ID/aims) or be pared down; currently it reads like an efficacy suggestion without context. Same for supplements—frame as personal choices; avoid implying disease-modifying effects. Add a one-sentence safety note (interactions/side-effects) to model responsible patient-authored guidance.

Pseudobulbar affect & sialorrhea

NS-GKT Comments: Dear Reviewer 3, thank you for reviewing the manuscript. I appreciate the time and effort you took to critique it thoughtfully.

This is a true story written by the patient about her conditions and progression. We cannot go deep into her clinical condition. We believe this is out of the scope of this manuscript.

Great to include lived tips (bite cloth). Consider adding 1–2 clinical pointers (“discuss with your care team …”) and avoid prescriptive language.

Clinical trials & eligibility

NS-GKT Comments: Dear Reviewer 2, thank you for reviewing the manuscript. I appreciate the time and effort you took to critique it thoughtfully.

This is a true story written by the patient about her conditions and progression. We cannot go deep into her clinical condition. We believe this is out of the scope of this manuscript.

The scale is ALSFRS-R, not “ALSFR-R/ALSFR-R.” Please correct throughout. This is a high-salience error for reviewers.

NS-GKT Comments: Dear Reviewer 2, thank you for reviewing the manuscript. I appreciate the time and effort you took to critique it thoughtfully.

This was an accidental typo error. Thank you for spotting this typo. We have corrected it.

The critique of strict eligibility is reasonable, but it needs either citations (platform trials, expanded access, adaptive designs) or reframing as a personal wish. A suggested constructive line: “Broader, phase-appropriate inclusion or pragmatic trials could enable participation across stages while still preserving safety and interpretability.”

NS-GKT Comments: Dear Reviewer 2, thank you for reviewing the manuscript. I appreciate the time and effort you took to critique it thoughtfully.

This is a true story written by the patient about her conditions and progression. We cannot go deep into her clinical condition. We believe this is out of the scope of this manuscript.

“I can easily tell if a drug is working” invites reviewer pushback (placebo, regression to mean). Soften: “I’m highly attuned to changes and would value contributing patient-reported outcomes to trials.”

Practical tips (eyes/lashes, laryngospasm)

NS-GKT Comments: Dear Reviewer 2, thank you for reviewing the manuscript. I appreciate the time and effort you took to critique it thoughtfully.

This is a true story written by the patient about her conditions and progression. We cannot go deep into her clinical condition. We believe this is out of the scope of this manuscript.

These are useful. To avoid over-generalization: add “This reflects my experience; please consult clinicians for diagnosis/treatment.”

“Laryngeal spasm” → “laryngospasm” is the more common clinical term; add a brief, neutral description. If you keep it, a single citation would go a long way; if not adding refs, explicitly flag as experience-based.

Nutrition/hydration & fasting

NS-GKT Comments: Dear Reviewer 2, thank you for reviewing the manuscript. I appreciate the time and effort you took to critique it thoughtfully.

This is a true story written by the patient about her conditions and progression. We cannot go deep into her clinical condition. We believe this is out of the scope of this manuscript.

The nutrition section reads prescriptive (“Don’t be afraid of a feeding tube”; “eliminated all unnatural sugars”). Reframe as “what worked for me,” and encourage dietitian involvement. Consider trimming brand callouts (e.g., “Häagen-Dazs”) to avoid perceived product placement.

NS-GKT Comments: Dear Reviewer 2, thank you for reviewing the manuscript. I appreciate the time and effort you took to critique it thoughtfully.

Furthermore, we acknowledge that support for ALS patients varies significantly across countries due to differences in healthcare systems, funding, and cultural factors.

Statements implying fatigue reduction or mucus changes from diet should be hedged unless you will cite. Otherwise: “I felt less tired when…” keeps it appropriately subjective.

Health-system reflections (Canada/Manitoba)

NS-GKT Comments: Dear Reviewer 2, thank you for reviewing the manuscript. I appreciate the time and effort you took to critique it thoughtfully.

This is a true story written by the patient about her conditions and progression. We cannot go deep into her clinical condition. We believe this is out of the scope of this manuscript.

Excellent contextualization, but claims about “most jobs” and “most cities” should be narrowed (“In my experience in Manitoba…”) or supported. If you keep the national view, add a qualifier noting variability by province/insurer.

A single sentence acknowledging that supports vary across regions and socioeconomic contexts will pre-empt reviewer critiques about representativeness.

NS-GKT Comments: Dear Reviewer 2, thank you for reviewing the manuscript. I appreciate the time and effort you took to critique it thoughtfully. We believe that this your suggestions are excellent. Therefore, we added the acknowledgment section in the manuscript.

Acknowledgments: NS and GKT appreciate the ALS patient community and patient-centric ALS organizations worldwide, which have motivated and inspired us to write this story. Furthermore, we acknowledge that support for ALS patients varies significantly across countries due to differences in healthcare systems, funding, and cultural factors.

References

Only four citations for wide-ranging claims is light, especially for the early context (Ice Bucket Challenge impact; advocacy and regulatory timeline; genetics at diagnosis; clinical-trial eligibility patterns; laryngospasm in ALS). Either (a) add 4–8 focused references, or (b) explicitly mark those statements as personal observations.

NS-GKT Comments: Dear Reviewer 2, thank you for reviewing the manuscript. I appreciate the time and effort you took to critique it thoughtfully.

NS-GKT Comments: Dear Reviewer 2, thank you for reviewing the manuscript. I appreciate the time and effort you took to critique it thoughtfully. We have accepted your recommendations and suggestions, and therefore, we have now increased the citations from 4 to 12. These papers cited in the manuscript are in line with the aims and objectives of the manuscript. Further, we have added the Aims and purpose of the study with conclusions.

Ensure uniform journal style (punctuation, author initials, capitalization, DOIs). Verify “ALSFRS-R” appears correctly wherever the scale is mentioned (also in any reference titles if relevant).

Formatting & style housekeeping

NS-GKT Comments: Dear Reviewer 2, thank you for reviewing the manuscript. I appreciate the time and effort you took to critique it thoughtfully.

We have accepted your recommendations and suggestions throughout the manuscript.

Standardize dashes (en-dashes for number ranges), units, and capitalization (e.g., “Magnetic Resonance Imaging (MRI)” then “MRI” thereafter).

Spell out acronyms on first use (ALSFRS-R; AIDS already OK; EMG/MRI OK).

NS-GKT Comments: Dear Reviewer 2, thank you for reviewing the manuscript. I appreciate the time and effort you took to critique it thoughtfully.

We have now expanded all the abbreviations suggested throughout the manuscript.

Tighten long sentences and remove colloquialisms that might read as prescriptive medical advice; keep the strong patient voice while adding light hedges (“in my case,” “I found”).

NS-GKT Comments: Dear Reviewer 2, thank you for reviewing the manuscript. I appreciate the time and effort you took to critique it thoughtfully.

Reviewer 3 Report

Comments and Suggestions for Authors

Amyotrophic lateral sclerosis (ALS) is a progressive and fatal neurodegenerative disorder that targets motor neurons in the brain and spinal cord. Although significant progress has been made in understanding its underlying mechanisms, effective treatments are still limited, and existing therapies primarily focus on slowing disease progression and enhancing patients’ quality of life. This manuscript presents a compelling first-person account of an ALS patient’s journey from the onset of symptoms to diagnosis and long-term management within the Canadian healthcare system. It offers meaningful insights into the lived experience of ALS, emphasizing important gaps in genetic testing, clinical trial eligibility, and practical aspects of patient care. The narrative is well-written, emotionally engaging, and makes a valuable contribution to patient-centered research and community awareness in ALS. With minor structural and editorial refinements—particularly to improve organization and strengthen connections to current literature—this manuscript would make an excellent addition to Healthcare.

Comments:

The introduction would be strengthened by a clearer and more structured framework. For instance, the authors should explicitly state the main aim and contribution of this perspective piece—whether it is to highlight diagnostic challenges, advocate for genetic testing, showcase patient resilience, or address all these aspects. Providing this clarification will help guide the reader and establish the context of the narrative more effectively.

While the manuscript includes several important references on ALS, these sources are not fully integrated into the narrative. For instance, discussions on diagnostic delays, clinical trial eligibility, and challenges in genetic testing could be better supported by incorporating recent literature, thereby enhancing the manuscript’s connection to current clinical and research contexts.

The narrative is engaging but somewhat lengthy and dense. It would benefit from being organized into distinct thematic sections—for example, early symptoms and diagnostic journey, genetic testing experience, treatment and disease management, and related topics—to improve readability and flow.

The “Highlights” section is valuable; however, including a discussion or concluding paragraph that reinforces these key messages would help ensure that clinicians, researchers, and policymakers can derive clear and actionable insights from the manuscript.

Author Response

Reviewer 3

Does the introduction provide sufficient background and include all relevant references?

Can be improved

Are all figures and tables clear and well-presented?

Yes

Comments and Suggestions for Authors

Amyotrophic lateral sclerosis (ALS) is a progressive and fatal neurodegenerative disorder that targets motor neurons in the brain and spinal cord. Although significant progress has been made in understanding its underlying mechanisms, effective treatments are still limited, and existing therapies primarily focus on slowing disease progression and enhancing patients’ quality of life. This manuscript presents a compelling first-person account of an ALS patient’s journey from the onset of symptoms to diagnosis and long-term management within the Canadian healthcare system. It offers meaningful insights into the lived experience of ALS, emphasizing important gaps in genetic testing, clinical trial eligibility, and practical aspects of patient care. The narrative is well-written, emotionally engaging, and makes a valuable contribution to patient-centered research and community awareness in ALS. With minor structural and editorial refinements—particularly to improve organization and strengthen connections to current literature—this manuscript would make an excellent addition to Healthcare.

NS-GKT Comments: Dear Reviewer 3, thank you for reviewing the manuscript. I appreciate the time and effort you took to critique it thoughtfully. Additionally, I want to thank you for your kind words about the manuscript. This is the first time our group under Dr. Tranmer has decided to publish the patient’s perspective. In addition to our research, our group is actively advocating for and engaging with ALS patients in the community. Our primary intention with this patient’s story is to create “HOPE” in the lives of those suffering from ALS. Furthermore, we aim to raise awareness and strongly advocate for ALS, a highly complex and incurable motor neurodegenerative disease.

The introduction would be strengthened by a clearer and more structured framework. For instance, the authors should explicitly state the main aim and contribution of this perspective piece—whether it is to highlight diagnostic challenges, advocate for genetic testing, showcase patient resilience, or address all these aspects. Providing this clarification will help guide the reader and establish the context of the narrative more effectively.

NS-GKT Comments: Dear Reviewer 3, thank you for reviewing the manuscript. I appreciate the time and effort you took to critique it thoughtfully. We have now included an introduction with the Aim and Purpose of this Story. Further, according to your suggestions, we have included highly cited journals in this section. We have now increased the citations from 4 to 11. The section below with all the newly added citations has been added to the manuscript in a tracked change.

ALS is a highly complex, multifactorial, idiopathic, heterogeneous, and incurable rare motor neurodegenerative disease. People worldwide are conducting research to slow the progression and improve the quality of life for ALS patients. For years, researchers have been making every effort to push the drug discovery engine to find a silver bullet molecule that could potentially slow disease progression. However, even 150 years after Charcot's discovery of ALS and 30 years after the identification of SOD1 mutations, progress in discovering and developing new treatments remains limited. Nevertheless, ALS patients and caregivers continuously inspire and motivate both the ALS and scientific communities. Despite suffering from ALS, patients continue to fight for the community and to raise funds for research worldwide. Additionally, ALS is a heterogeneous disease, with each patient experiencing different rates of progression, which makes understanding it highly complex [5]. Many ALS patients may lose hope due to the disease's progressive and debilitating nature, which is caused by several factors. First, diagnostic delays often mean patients receive a diagnosis too late, sometimes because the disease progresses rapidly[6]. Overlapping symptoms with other neurological disorders can further delay diagnosis. Second, resource scarcity poses a challenge, including financial burdens from expensive treatments, limited access to assistive devices like accessible vans, inadequate in-home care, and gaps in caregiver support[7]. Third, navigating clinical trials is difficult for ALS patients due to logistical barriers, strict eligibility criteria, and informational challenges[8]. Fourth, barriers to genetic testing include inconsistent access, high costs, limited genetic counseling, and provider awareness issues[9]. Lastly, many patients lose hope in the drug discovery process itself, as most clinical trials have failed to demonstrate efficacy, potentially due to the disease's complexity and heterogeneous nature, as well as ill-defined clinical endpoints and poorly designed clinical trials[10]. However, we firmly believe that the drug discovery pipeline for ALS should involve a broader scientific community, including patients, basic scientists, clinicians, and medicinal chemists. Currently, medicinal chemists are underrepresented in the discovery and development of ALS drugs. They develop drugs by designing, synthesizing, and optimizing chemical compounds to create new medicines[11].

We believe sharing patient stories can raise awareness about the disease and foster strong advocacy within the ALS community for specialized, holistic, multidisciplinary care[12]. Sherry’s story is unique.

Sherry’s journey from weakness to diagnosis highlights the diagnostic challenges, advocates for genetic testing, and showcases her resilience, while also emphasizing the importance of flexible clinical trial inclusion criteria. Through her story, Sherry remains highly optimistic about improving the lives of ALS patients by raising awareness, encouraging resilience, perseverance, and fostering hope. Her story aims to influence the perspectives of ALS patients and inspire change positively. Additionally, her story aims to motivate, inspire, and encourage ALS patients worldwide to develop a positive will to live despite the disease.

While the manuscript includes several important references on ALS, these sources are not fully integrated into the narrative. For instance, discussions on diagnostic delays, clinical trial eligibility, and challenges in genetic testing could be better supported by incorporating recent literature, thereby enhancing the manuscript’s connection to current clinical and research contexts.

NS-GKT Comments: Dear Reviewer 3, thank you for reviewing the manuscript. I appreciate the time and effort you took to critique it thoughtfully. We have now included an introduction with the Aim and Purpose of this Story. Further, according to your suggestions, we have included highly cited journals in this section. We have now increased the citations from 4 to 11. The section below with all the newly added citations has been added to the manuscript in a tracked change.

The narrative is engaging but somewhat lengthy and dense. It would benefit from being organized into distinct thematic sections—for example, early symptoms and diagnostic journey, genetic testing experience, treatment and disease management, and related topics—to improve readability and flow.

NS-GKT Comments: Dear Reviewer 3, thank you for reviewing the manuscript. I appreciate the time and effort you took to critique it thoughtfully. This is excellent advice to make it more organized. We accepted your suggestions and developed a flow schematic of Sherry’s journey for better readability.

we have introduced a new Figure 2, which illustrates various Stages of my Journey from Symptoms of Weakness to Diagnosis of Amyotrophic Lateral Sclerosis (ALS). This figure is unique, and we tried to simplify the journey of Mrs. Sherry through this figure, mentioning the journey in different stages. Please refer to Figure 2, which is included in the manuscript. This will provide a better organization of her trip, enhancing readability for the readers.

The “Highlights” section is valuable; however, including a discussion or concluding paragraph that reinforces these key messages would help ensure that clinicians, researchers, and policymakers can derive clear and actionable insights from the manuscript.

NS-GKT Comments: Dear Reviewer 3, thank you for reviewing the manuscript. I appreciate the time and effort you took to critique it thoughtfully. You have given an excellent suggestions and recommendations to include the conclusive statement.

We tried to do our best according to the patient perspective. Please see below we have included conclusions and Final Words for the Policy Makers in the Government:

Conclusions:

Sherry’s conclusion and wish, conveyed through her story of resilience, is to establish an ALS research hub dedicated to improving the early diagnosis of the disease. This hub should include genetic counseling and testing for all patients, regardless of their diagnosis, and encompass their family members as well. Additionally, she advocates for a centralized, user-friendly knowledge portal for ALS patients and caregivers, providing current information on ongoing clinical trials, with clear navigation and criteria. She also urges broader inclusion criteria for clinical trial participation, emphasizing that ALS, being a rare and incurable disease, warrants that all patients should have the right to participate, regardless of disease stage, genetic makeup, or heterogeneity. She believes every patient’s contribution can advance drug discovery and development. Furthermore, Sherry emphasizes the importance of diverse research teams, including clinicians, scientists, medicinal chemists, patients, and caregivers, to incorporate multiple perspectives in the drug discovery process. Through her story, she aims to advocate for increased government funding for ALS research.

Final Words for the Policy Makers in the Government:

Rare diseases, such as ALS, can often become neglected diseases, as the numbers of patients seem small, but the need remains high. Policy makers in the government need to focus on the high need of ALS patients for inclusion in the latest research. Serving the high needs of one ALS patient serves the needs of many, including caregivers, the community, and practitioners.
